# Beneficial Effect of Methanolic Extract of Frankincense (*Boswellia Sacra*) on Testis Mediated through Suppression of Oxidative Stress and Apoptosis

**DOI:** 10.3390/molecules27154699

**Published:** 2022-07-22

**Authors:** Samir Abdulkarim Alharbi, Mohammed Asad, Kamal Eldin Ahmed Abdelsalam, Monjid Ahmed Ibrahim, Sunil Chandy

**Affiliations:** Department of Clinical Laboratory Science, College of Applied Medical Sciences, Shaqra University, Shaqra P.O. Box 1383, Saudi Arabia; saalharbi@su.edu.sa (S.A.A.); kabdelsalam@su.edu.sa (K.E.A.A.); monjid@su.edu.sa (M.A.I.); schandy@su.edu.sa (S.C.)

**Keywords:** Bcl-2, boswellic acids, caspase-3, sperm count, sperm motility

## Abstract

*Boswellia sacra* oleo gum resin (*Burseraceae*) commonly known as frankincense is traditionally used in many countries for its beneficial effect on male fertility. This study explores its effect on the male reproductive system after a 60-day repeated administration at two different doses to rats (in vivo) and on human Leydig cells (in vitro). The methanolic extract of *B. sacra* was analyzed for the presence of various constituents by preliminary phytochemical analysis and gas chromatography-mass spectrometry (GC-MS) while quantitative analysis of boswellic acids was done by high-performance liquid chromatography (HPLC). Administration of *B. sacra* extract to rats elevated the serum testosterone levels with an associated reduction in serum levels of FSH and LH. An increase in the activity of antioxidant enzymes, superoxide dismutase and catalase, was seen. A dose-dependent increase in the sperm count and sperm motility was also observed. The in vivo results were supported by changes in the expression of the *Bcl-2* gene and *caspase-3* gene in human Leydig cells in vitro. The results of this study support the traditional use of *B. sacra* to increase male fertility.

## 1. Introduction

Male infertility is one of the disorders that is treated using traditional medicines. Several herbal drugs and alternative medicines are available in different countries that are believed to promote fertility and/or prevent infertility in males. This has led to the availability of herbal and nutritional supplements as fertility promoters in different parts of the world [1,2,3]. The pharmacotherapy of infertility involves the administration of testosterone, nutritional supplements, and antioxidants to increase sperm count and sperm motility [4].

One of the traditional medicines that is believed to promote fertility in men is frankincense, which is an oleo gum resin from the plant *Boswellia sacra* Fluck. belonging to the family Burseraceae. It is one of the most popular herbal medicines that is used for several purposes ranging from chewing as a mouth freshener, and perfumery during religious rituals to the treatment of many diseases. In traditional medicine, *Boswellia sacra* oleo gum resin (*B. sacra*) is used for the treatment of digestive, skin, ear, and throat infections, relief of menstrual pain, cardiovascular and neurologic problems. One of the reasons for its popularity is that it is mentioned in holy books and other ancient textbooks. Moreover, products made from *Boswellia* oleo gum resin are marketed throughout the world for various biological activities [5,6].

Our lab is involved in the evaluation of *B. sacra* for pharmacological and toxicological effects [7,8,9]. An early study carried out in our lab with the water extract of *B. sacra* in rats showed that it increases gastric acid secretion in rats and exacerbates gastric ulcer formation in experimental gastric ulcer models [7]. However, a 28-day repeated dose study for the evaluation of other organ toxic effects using standardized methanolic extract of *B. sacra* showed that it is devoid of any hepatotoxicity or nephrotoxicity [8]. A further 28-day repeated dose study to determine the effect on the male reproductive system in rats revealed that *B. sacra* methanolic extract increases the expression of *GSTPi, IGFBP3*, and *HSP70* genes in testis and elevates serum testosterone levels indicating that it may have a protective effect on the testis [9].

The *Boswellia* species are used traditionally in Jordan as an aphrodisiac and fertility promoter [10] while a traditional medicine containing frankincense along with other herbs is used in Iran to treat infertility in men [11]. An earlier study on the effect of another *Boswellia* species *Boswellia thurifera* oleo gum resin reported that it increases spermatogenesis and sperm motility in rats [12]. The current study was envisaged by keeping in view the results of our previous study on the effect of *B. sacra* in male rats and the traditional uses of *Boswellia* species as fertility promoters in different cultures. An in vivo 60-day repeated oral administration study was performed to determine the effect of *B. sacra* on sperms, testes, antioxidant enzymes, and serum hormonal levels in rats along with an in vitro study on the human Leydig to determine cytotoxic effect and expression of apoptotic genes.

## 2. Results

### 2.1. Analysis of the Extract

Preliminary phytochemical analysis of the extract showed the presence of different constituents such as alkaloids, flavonoids, terpenoids, tannins, and phenols. The high-performance liquid chromatography (HPLC) analysis for quantification of boswellic acids revealed that the total amount of boswellic acids was around 37% *w*/*w* of the extract. Out of this, around 21% was boswellic acid (α + β), 7.5% was acetyl boswellic acid (α + β) followed by 11-keto-β-boswellic acid (6.23%) and acetyl-11-keto-β-boswellic acid was present in the least amount (1.89%). The HPLC chromatogram is shown in Figure 1. The gas chromatography-mass spectrometry (GC-MS) chromatogram showed the presence of 71 different constituents (Figure 2) and the major constituents revealed by GC-MS are listed in Table 1. The list of all compounds is given as Appendix A.

### 2.2. Effect on Testes in Rats

There was no significant change in the weight of the animals during the treatment period. The overall gain in the weight at the end of the treatment period in animals receiving either dose of *B. sacra* extract was not significantly different from the control group. Similarly, the weight of the testis and the relative weight of the testis to body weight (%) were not significantly different in the treatment group compared to the control group (Figure 3). No significant difference in the weight of cauda epididymis was observed. The data are presented as Appendix A.

A significant increase in serum total testosterone was observed in a dose-dependent manner after administration of *B. sacra* extract (*p* < 0.001). This was associated with a negative feedback-induced decrease in the serum levels of FSH and LH (Figure 4).

A dose-dependent increase in sperm count was observed after administration of *B. sacra* extract (*p* < 0.01). A few sperm defects were observed in the animals such as straight heads, headless sperms, banana heads, different types of tail bends, and a few cytoplasmic residues. The sperm defects (%) were not significantly different in the treated groups when compared to the control (Figure 5). The percentage of sperms with progressive motility was significantly increased by the extract with a significant decrease in the non-progressive and immotile sperms when compared to control (Figure 6).

The activities of antioxidant enzymes—SOD and catalase were significantly enhanced by the extract (Figure 6).

The effect of different treatments on the seminiferous tubules is shown in Figure 7. The cytoarchitecture of the testis was well maintained after *B. sacra* extract administration with no indication of any edema or damage to the spermatocytes or the sperms. The number of primary spermatocytes, secondary spermatocytes, spermatids, and sperms was more in the seminiferous tubules in the treated animals compared to control. However, no noticeable change in the size of seminiferous tubules was observed.

### 2.3. Effect on Human Leydig Cells (In Vitro)

The *B. sacra* extract did not affect the cell viability of human Leydig cells as shown in Table 2. About 98% cell viability was observed up to a concentration of 100 µg/mL while LPS, a standard cytotoxic agent, reduced cell viability in a concentration-dependent manner (Figure 8). A significant increase in the expression of the *Bcl-2* gene with a significant decrease in the expression *caspase-3* gene was observed in a concentration-dependent manner after incubation of cells with *B. sacra* extract (Figure 9).

## 3. Discussion

*Boswellia* is one of the popular herbal medicines that is traditionally used and is also marketed throughout the world. It is used widely for the treatment of arthritis and inflammation apart from its use in the treatment of other diseases due to its potent antioxidant action. The current study was undertaken to ascertain the traditional belief about its beneficial effect on the reproductive system. The *Boswellia* species used in different countries varies and depends on the availability of a particular species in that region, but all these different species are called “frankincense”. The active constituents in all the species are similar [13]. Many of the reported activities of *Boswellia* are due to the presence of boswellic acids [14,15]. Apart from boswellic acids, *Boswellia* is rich in volatile constituents and these volatile components are also reported for antitumor, anti-inflammatory, and antioxidant actions [16]. *B. serrata* is used widely in South Asian countries. In the Middle Eastern and Arab-African countries, *B. serrata* and *B. sacra* are available. Of these two in the Middle East, *B. serrata* is considered inferior in quality while *B. sacra* (called *Omani Luban*) is popular and expensive because it is considered to be of better quality than *B. serrata*. Hence, the current study was carried out using *B. sacra*. The preliminary phytochemical analysis was performed to detect the presence of different classes of constituents and the GC-MS analysis was carried out to determine the presence of volatile constituents. The HPLC analysis was done to standardize the extract for the amount of boswellic acids, which are believed to be the main chemical constituents responsible for the pharmacological effects of *Boswellia* species [14]. The different phytochemical constituents revealed by GC-MS analysis will be explored in the future to identify the component(s) responsible for the observed beneficial effect on the testis. The use of extract instead of pure boswellic acid and/or pure volatile components revealed an overall effect of the oleo gum resin to support the traditional claim.

As mentioned earlier, frankincense is a popular traditional drug for the treatment of male infertility in different countries. Despite these traditional claims for beneficial effects of *Boswellia*, a toxic effect on the rat testis was reported after inhalation of “*smoke* “by burning of *Boswellia* species such as *B. papyrifera* and *B. carterii* [17]. On the contrary, in our earlier study, the methanolic extract of *B. sacra* decreased the expression of *GSTPi, IGFBP3,* and *HSP70* genes with no noticeable changes in the cytoarchitecture of rat testis after a 28-day repeated dose administration indicating that it may protect the testis against toxicity [9].

The present study is a continuation of our earlier study by extending the treatment period to 60 days and beginning the treatment at an early age of the animals. Young animals aged 28 to 30 days were selected as it is known that agents targeting the male reproductive system will show a more profound effect if the administration is started at a young age and continued for a long duration [18]. The doses of the extract were selected based on an acute toxicity study carried in our earlier studies [8,9]. The extract administered orally was safe up to a dose of 2000 mg/kg and based on this, moderate doses of 250 mg/kg and 500 mg/kg were selected. The acute toxicity study was carried out as per the OECD guidelines following the Up and Down method [19]. Sperm parameters such as sperm count, sperm motility and sperm defects, serum hormonal levels of testosterone and gonadotrophins along with antioxidant enzyme levels, and proof of spermatogenesis in the testis were studied.

The weight of the animals, testis weight, and relative weight of testis to body weight were not affected by the *B. sacra*. These parameters are used to indicate the overall effect of chemicals and drugs on the male reproductive system. Agents that induce toxic effects on the testis such as anticancer drugs produce a decrease in the body weight and the weight of reproductive organs [20] while those that produce a beneficial effect are reported to either maintain the body weight and testicular weight or induce an increase in their weights [21,22].

The antioxidant defense mechanisms in the testis are weak due to the limited amount of antioxidant enzymes and the generation of reactive oxidative radicals from enzymes such as NADPH oxidase, xanthine oxidase, and mitochondrial electron transport chain [23]. Antioxidants are known to protect the testis and several antioxidants are reported to prevent testicular damage [24,25]. Furthermore, antioxidants are also recommended for the treatment of infertility to protect testis from endogenous oxidants and it has been reported that consumption of antioxidant supplements by men may increase the live birth rate in infertile couples [26]. Nutraceuticals are being increasingly recommended to prevent testicular injury due to toxicants [27]. *B. sacra* is a well-known antioxidant and it is known to protect several organs against oxidative damage both in vivo and in vitro [28,29]. In the current study, it increased the levels of antioxidant enzymes, SOD and catalase, indicating that it may protect the testis against oxidative damage. As mentioned above, in our earlier study, *B. sacra* extract had decreased the expression of genes involved in the synthesis of antioxidant enzymes in normal animals, suggesting that it reduces the level of reactive oxidative species in the testis [9]. Since *Boswellia* species are known to have antioxidant effects in vitro, the increase in the activities of the antioxidant enzymes may at least be partly due to direct scavenging of endogenously generated reactive oxidative species by the extract.

Sperm count, sperm motility, and sperm defects are the common parameters used to detect male infertility. Several causes and factors have been identified for the reduction in sperm count, sperm motility, and morphological changes in the sperm including genetics and oxidative stress [30]. Oxidative stress is considered one of the main mechanisms contributing to sperm defects [31]. Generation of oxygen free radicals leads to peroxidation of lipids, oxidation of sulfur-containing proteins, damage to DNA, and oxidative stress in mitochondria leading to reduced availability of ATP [32]. The sperm motility is impaired due to the reduced amount of ATP [33]. *B. sacra* increased the sperm count and sperm motility. The effect on sperm motility and number were supported by histological studies on the testis, where the density of spermatocytes and sperms was more in *B. sacra* treated rats compared to the control. The effects on sperms could be due to the antioxidant action of *B. sacra* extract. This antioxidant action may protect testis against oxidative damage that in turn leads to an increase in sperm production while the effect on the sperm motility could be due to enhanced availability of the ATP due to scavenging of oxidative species in the mitochondria.

The serum testosterone level was increased after *B. sacra* treatment. Testosterone synthesis is regulated in the testis through activities of enzymes such as 3β-hydroxystreroid dehydrogenase and 17β-hydroxysteroid dehydrogenase, aromatase, cholesterol side-chain cleavage enzyme-P450cc [34]. The increase in the serum total testosterone levels in the present study is consistent with the results of our earlier study, where a similar effect was observed after 28-day repeated administration [9]. However, an additional study on the effect of *B. sacra* on the secretion of gonadotrophins, FSH and LH, was included. The *B. sacra* extract did not increase the secretion of these gonadotrophins, suggesting that the increase in the levels of serum total testosterone could be due to the antioxidant effect and stimulation of steroid synthesis in the testis. An expected decrease in the serum levels of FSH and LH was observed due to physiological feedback regulation. Testosterone synthesis is stimulated by the LH and an increase in the testosterone secretion stimulates a negative feedback mechanism to prevent the release of gonadotrophin-releasing hormone from the hypothalamus which in turn decreases the release of FSH and LH from the pituitary gland [35].

The current study was done using young animals as the extract had shown effect in adult rats in our previous study after 28-day repeated administration [9]. Since young animals were used, it can be suggested that the extract might have also influenced the development of the reproductive system. Further studies on adult rats after 60-days repeated administration and the comparison of results from young animals with adult animals may help in understanding the effect of *B. sacra* extract on reproductive development. The in vivo study was done using only two doses. Evaluation of effect at a wide range of doses will help in determining the threshold dose and maximum effective dose of the extract.

The in vitro study determined the effect of *B. sacra* on the human testicular cells. This study also determined if the active constituent(s) involved in protection against the testicular damage acts directly or through their metabolites. The Leydig cells were unstimulated and the study was done to determine the effect on the expression of genes involved in apoptosis. The results showed that *B. sacra* extract is safe and it maintained the viability of human Leydig cells in concentrations up to 100 µg/mL (98%). LPS is an endotoxin that is used to simulate oxidative testicular damage and this was used as a standard agent to determine cytotoxicity [36,37]. LPS induced an increase in the expression of *caspase-3* and a decrease in the expression of the *Bcl-2* gene while an opposite effect was observed after incubation of human Leydig cells with *B. sacra* extract. An increase in the expression of *Bcl-2* and a reduction in *caspase-3* expression might have contributed to an increase in catalase activity observed in vivo in the rat testes. *Bcl-2* is known to decrease hydrogen peroxide formation while *caspase-3* is reported to increase hydrogen peroxide levels. The increase in catalase activity could be due to reduced hydrogen peroxide levels [38]. However, earlier reports suggest that *Bcl-2* does not change the activity of SOD [39] while an increase in catalase activity associated with a reduced expression of *caspase-3* has been reported in many tissues [40,41]. On the contrary, administration of exogenous SOD is reported to induce expression of *caspase-3* in prostate cancer cells lines [42].

The findings of the current study can be further confirmed by studying the effect of *Boswellia sacra* oleo gum resin extract on rat infertility models such as cyclophosphamide-induced testicular toxicity or cisplatin-induced testicular toxicity in rats.

## 4. Materials and Methods

### 4.1. Chemicals

The reagents, chemicals, and diagnostic kits were procured from different suppliers and they are mentioned below.

### 4.2. Preparation of Methanolic Extract of B. sacra

Frankincense locally called “*Omani Luban”* was purchased from the local market. The ole gum resin was identified by a botanist through an earlier voucher specimen kept in the institute (SU/CAMS/3/2018). The powdered oleo gum resin was extracted using 90% methanol (SD Fine Chemicals, Mumbai, India) following the standard procedure of Soxhlation [43]. The extraction yield was 13.2% *w*/*w* of the oleo gum resin. The extract was suspended in water using sodium carboxymethylcellulose (1% *w*/*v*) (LobaChemie, Mumbai, India).

### 4.3. Analysis of the Extract

The extract was subjected to preliminary phytochemical analysis [44] followed by the GC-MS analysis for the identification of different constituents. The boswellic acid content was determined by the HPLC method.

#### 4.3.1. GC-MS Analysis

A GC-17A gas chromatograph (Shimadzu, Coppell, TX, USA) attached to a GC-MS QP 5050A Mass Spectrometer (Shimadzu) with a GC-MS electron impact ionization method was used. A fused silica capillary column (DB-5 ms-J&W) with dimensions of 30 m × 2.5 mm, 0.25 mm was employed. The conditions for the GC-MS run are as follows: the injection temperature and the interface temperature were kept at 300 °C while the ion source was adjusted to 250 °C, with helium as the carrier gas at a flow rate of 1 mL/min. Isothermal heating at 100 °C (1 min) followed by 300 °C for 20 min was used. The mass spectra were quantified by studying the peak areas and referring to the internal standard.

#### 4.3.2. Determination of Boswellic Acids by HPLC

The HPLC method has been described in detail earlier [8]. Methanol was used to dissolve the extract and standard boswellic acids. The dissolved constituents were injected into the chromatographic system (Shimadzu). A mixture of 950 mL water and 50 mL acetonitrile with 100% methanol was used as the mobile phase. The detector was set at 210 nm for the detection of boswellic acids (α and β) while the 11-keto-β-boswellic acid and acetyl-11-keto-β-boswellic acid were detected at 247 nm.

### 4.4. Animals

Male rats of Wistar strain aged 28 to 30 days that were maintained in the institutional animal house were used [45]. Animals were maintained under controlled temperature and humidity with access to rat chow (VerseleLaga, Deinze, Belgium) and water. The experimental protocol consisted of standard methods and procedures with minor modifications. The protocol was reviewed and approved by the Ethical Research Committee of Shaqra University (approval number—53/10315). The number of animals was selected based on sample size calculations. All the biochemical, physiological, and histopathological examinations were carried out by researchers who were blind to the treatment.

### 4.5. Effect on the Male Reproductive System in Rats

Three groups of rats consisting of six animals each were treated as follows: the first group received vehicle, sodium carboxymethylcellulose (LobaChemie, Mumbai, India), 1% *w*/*v* for 60 days and served as control. The second group and third group were treated with a suspension of methanolic extract of *B. sacra* orally at a dose of 250 mg/kg/day and 500 mg/kg/day for 60 days respectively [8]. The body weight of the animals was noted every 7 days throughout the treatment period.

On day 60, animals were anesthetized using a cocktail of ketamine (91 mg) (Hikma Pharmaceuticals, Riyadh, Saudi Arabia) and xylazine (9.1 mg) (InterchemieWerken, Venray, Netherlands) at a dose of 1 mL/kg intraperitoneally [46]. Blood was withdrawn through retro-orbital plexus and serum was used to determine total testosterone [47], FSH, and LH levels by ELISA [48]. The testes and cauda epididymis were isolated and weight was determined separately. The cauda epididymis was used to determine sperm parameters while the testes were subjected to estimation of antioxidant enzyme activities and histological analysis. The animals were euthanized by giving a further dose of ketamine and xylazine (five times the anesthetic dose) [46].

#### 4.5.1. Sperm Parameters

The isolated cauda epididymis was suspended in 3 mL of Hank’s buffered salt solution (HBSS) (Chemicals used were from SD fine chemicals and LobaChemie, Mumbai, India) [49]. The epididymis was minced in this solution using a scissor to take out the sperms into the HBSS. For determination of sperm motility, a drop of the above suspension on a slide was magnified under 100×. Two hundred sperms were observed. The numbers of progressive, non-progressive, and immotile sperms were counted as per the WHO specifications [50]. The fast progressive and slow progressive were categorized under a single group as progressive sperms [51]. The sperm count was estimated by diluting the sperm suspension with sodium bicarbonate-formalin diluting solution (Chemicals used were from SD fine chemicals and LobaChemie, Mumbai, India) (1:20) to stop the sperm motility [52]. A drop of this was inserted into Neubauer’s chamber (Aiken, CA, USA) and the numbers of sperms in two WBC chambers were counted under 100x magnification to determine the number of sperms/mm^3^ by multiplying the number counted by 100,000. The sperm morphology was studied by preparing a smear of the above suspension. A drop of suspension of cauda epididymis in HBSS was smeared on a clean slide. It was air-dried and then stained with eosin (0.5% *w*/*v*) (WINLAB, Leicestershire, UK) for 5 min followed by air drying [52]. The morphology was studied under 400× to determine defects in the head and tail. A total of 200 sperms were observed for the morphological defects in each animal. Sperm abnormalities of the mid-region were considered as part of tail abnormality [53].

#### 4.5.2. Testis Parameters

Antioxidant enzyme activities and histological studies were carried out. One testis from each animal was homogenized with 0.25% *w*/*v* sucrose (SD Fine chemicals, Mumbai, India) in phosphate buffer (all constituents were from either from LobaChemie, Mumbai, India or SD Fine chemicals, Mumbai, India) at pH 7.4. The superoxide dismutase (SOD) and catalase activities were determined by standard procedures [54,55].

The SOD activity was determined by homogenizing the tissue in 0.25% sucrose in phosphate buffer (pH 7.4) and the homogenate was centrifuged at 3000 rpm for 10 min. The supernatant was used for determination of SOD activity while for the estimation of catalase activity, tissue was homogenized in 0.15 M potassium chloride followed by centrifugation at 800 rpm for 10 min and the supernatant was collected to determine catalase activity. The SOD activity was measured by autooxidation of hydroxylamine that causes reduction of nitro blue tetrazolium (NBT) leading to nitrite production in the presence of ethylenediamine tetraacetic acid (EDTA). The reduction of NBT was measured at 560 nm spectrophotometry. Estimation of catalase was done by decomposition of hydrogen peroxide in an assay mixture containing phosphate buffer (pH 7) at 240 nm.

For histological studies, the left testis was fixed with 10% neutral formalin prepared from formalin solution (SD Fine chemicals, Mumbai, India) followed by dehydration and embedding in paraffin wax (LobaChemie, Mumbai, India). The cut sections were stained with hematoxylin (BDH Middleeast LLC, Riyadh, Saudi Arabia) and eosin stain (WINLAB, Leicestershire, UK). The seminiferous epithelium was examined to identify the effect on spermatogenesis and other morphological changes.

### 4.6. Effect on Human Leydig Cells (In Vitro)

#### 4.6.1. Cell Lines

Human Leydig cells (Cat. # 4510) were acquired from Scien Cell Research Labs (Carlsbad, CA, USA) and it was cultivated in cell culture medium-Leydig cell medium (Cat. # 4511, Scien Cell Research Labs, Carlsbad, CA, USA) with fetal bovine serum (HiMedia, Mumbai, India).

#### 4.6.2. MTT Viability Assay

Cell suspension (200 µL) was seeded into a 96-well plate (Corning, Glendale, AZ, USA) at a cell density of (20,000 cells per well) to grow for 24 h at 37 °C with 5% CO_2_ (Healforce, Shanghai, China). Lipopolysaccharide (LPS) (Sigma, Mumbai, India, Cat No. L4391) or *B. sacra* extract at different concentrations were added to these cells and incubated for another 24 h. After removing the spent media, 3-(4,5-dimethylthiazol-2-yl)-2,5-diphenyltetrazolium bromide (MTT) (HiMedia, Mumbai, India) reagent was added to adjust the final concentration to 0.5 mg/mL. The plates were protected from light and incubated for 3 h followed by removal of MTT and addition of 100 µL of dimethylsulfoxide (DMSO) (Sigma, #PHR1309); a solubilizing agent with gentle stirring and pipetting up and down to dissolve the MTT. The absorbance was read at 570 nm wavelength to determine cell viability [56].

#### 4.6.3. Effect on Apoptotic Markers

Cell suspension (2000 μL) was seeded in a 6-well plate (1 million cells per well). After 12 h growth, cells were treated with *Boswellia* extract for 24 h. After the removal of the spent medium, cells were collected by trypsinization.

The RNA was isolated using a Qiagen RNeasy kit (Qiagen Sciences, Germantown, MD, USA), treated with DNAse (to avoid genomic DNA contamination), and purified. The RNA was quantified by UV-Visible spectroscopy (Qiaexpert, Hombrechtikon, Switzerland) and cDNA was synthesized (Bioradiscript cDNA synthesis kit, Gurugram, India) using random hexamer + oligodT primers as per the following reaction; 5× Mix—5 µL, nuclease-free water—3 µL, RNA—2 µg in 15 µL, RT enzyme—2 µL. The primers used are given in Table 3. The cDNA cycle was as follows; priming—5 min at 25 °C, RT—20 min at 46 °C, RT inactivation—1 min at 95 °C. Reaction volume for real-time PCR was 25 µL (SYBR mix—12.5 µL, water—9.5 µL, Forward primer 1 µL, Reverse primer 1 µL, cDNA 1 µL). Real-time PCR was performed in a Rotor-gene machine (Qiagen). QuantiFast SYBR green master mix (Qiagen) was used and primers were validated with SYBR reactions for amplification and melt curves. The concentration of primers used in real-time PCR was 200 nM. No primer had self-annealing or self-dimerization property and they had Tm near 60 °C. Reactions were done at 60 °C, PCR initial activation step—5 min at 95 °C, denaturation at 95 °C—10 s, annealing at 60 °C—20 s, extension—20 s at 72 °C. The number of cycles performed was 40 and the average of duplicated reactions was taken for analysis. The ΔΔCt method was used for calculating fold changes.

### 4.7. Statistical Analysis

Values are expressed as mean ± SEM as mentioned in the footnotes. Statistical differences between the groups were analyzed using one-way ANOVA with Tukey’s post-test. The statistical analysis was done using SPSS for windows, version 22 (SPSS Inc., Chicago, IL, USA). *p* < 0.05 indicated statistically significant difference. The distribution of data was assessed by Skewness value. A skewness value between −1 and +1 was used to indicate normal distribution [57].

## 5. Conclusions

*B. sacra* (frankincense) extract showed a potential protective effect in the testis that is mediated at least in part due to its antioxidant action. It caused an increase in the sperm count and sperm motility along with an increase in spermatocytes in the seminiferous tubules. The extract also increased serum testosterone levels. The beneficial effect was further confirmed by in vitro studies wherein *B. sacra* extract did not affect cell viability in human Leydig cells and modulated the expression of apoptotic genes.

## Figures and Tables

**Figure 1 molecules-27-04699-f001:**
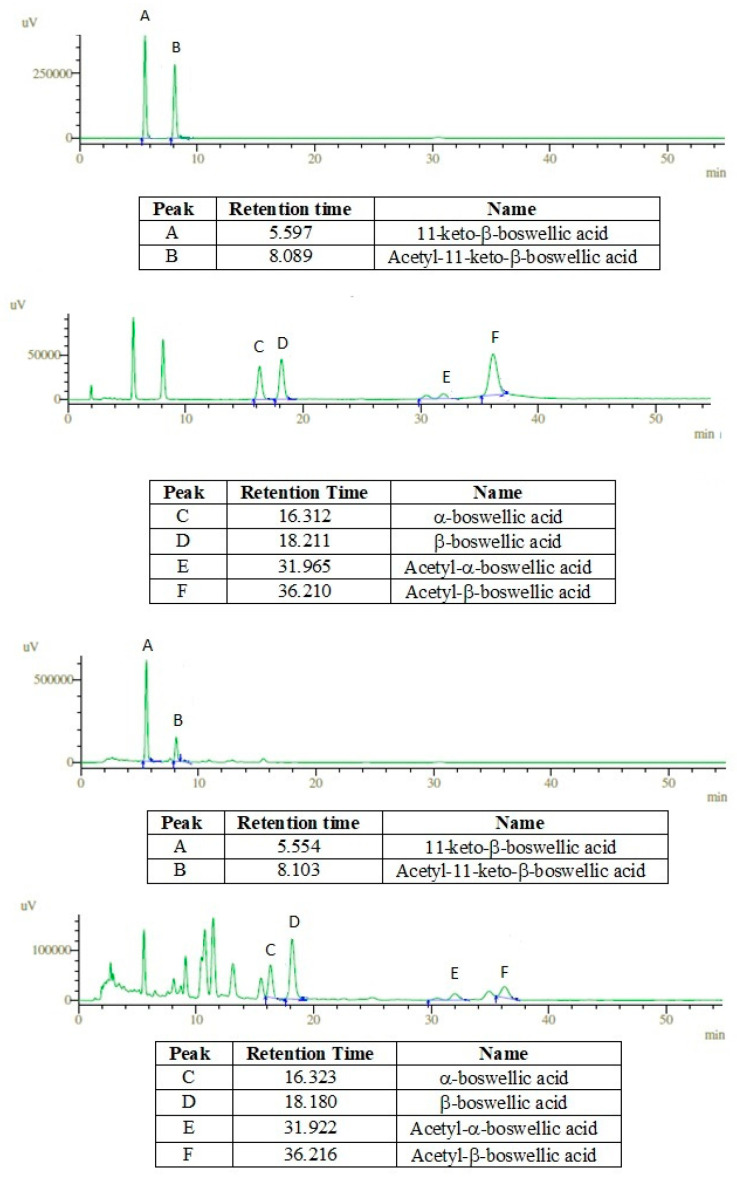
HPLC chromatogram showing peaks of standard boswellic acids mix (first two) and methanolic extract of *B. sacra* (last two).

**Figure 2 molecules-27-04699-f002:**
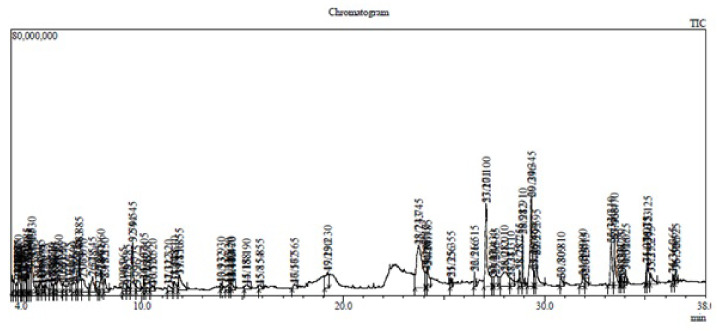
GC-MS chromatogram of methanolic extract of *B. sacra* showing different peaks.

**Figure 3 molecules-27-04699-f003:**
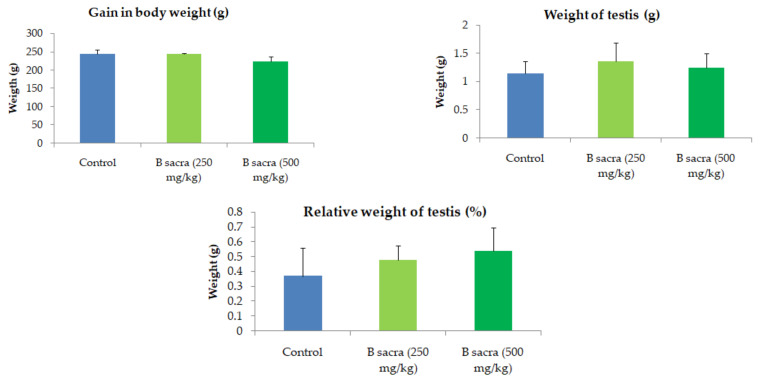
Effect on body weight, testis weight, and relative weight of testis to body weight. The gain in body weight indicates the difference in the weight of animals at the end of the treatment period and the beginning of the treatment. The weight of the testis was taken at the end of the treatment period while the relative weight of the testis shows the percentage of the testis in relation to the total body weight. One-way ANOVA followed by Tukey’s post-test was used for statistical analysis. All values are mean ± SEM, *n* = 6. There was no significant change in the weights between the control and *B sacra* groups.

**Figure 4 molecules-27-04699-f004:**
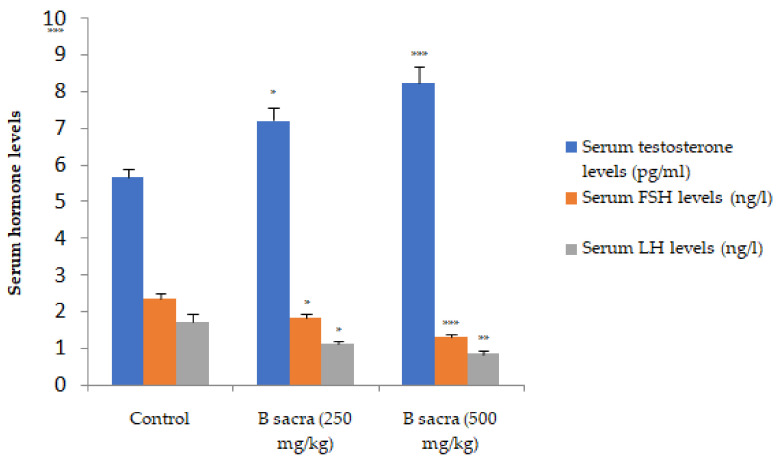
Effect on serum hormone levels. All the hormone levels were determined using ELISA at the end of the treatment period. Statistical significance was determined using one-way ANOVA followed by Tukey’s test. All values are mean ± SEM, *n* = 6. * *p* < 0.05, ** *p* < 0.01, *** *p* < 0.001 compared to control.

**Figure 5 molecules-27-04699-f005:**
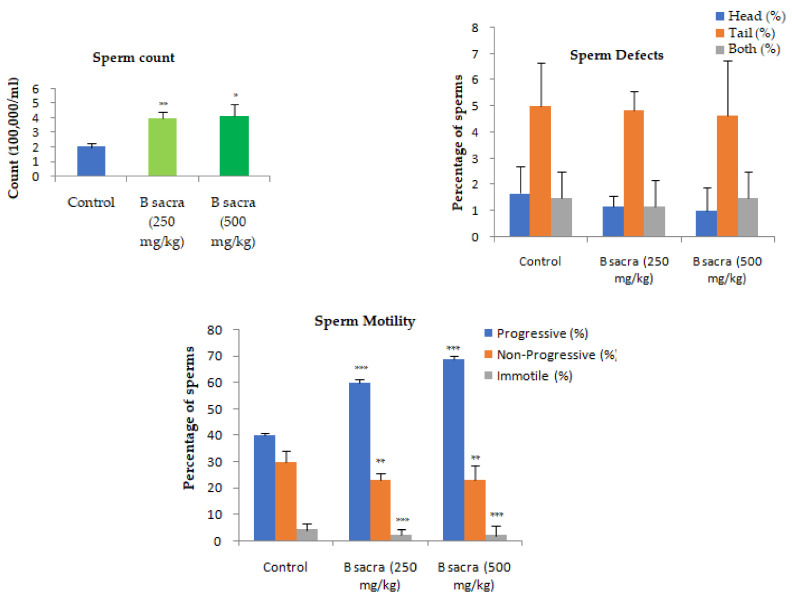
Effect on different sperm parameters. All the sperm parameters were measured in the cauda epididymis at the end of the treatment period. Sperm counts were measured using Neubauer’s chamber under 100×. Sperm defects were studied under 400× and sperm motility under 100×. Statistical analysis was done using one-way ANOVA followed by Tukey’s test. All values are mean ± SEM, *n* = 6. * *p* < 0.05, ** *p* < 0.01, *** *p* < 0.001 compared to control.

**Figure 6 molecules-27-04699-f006:**
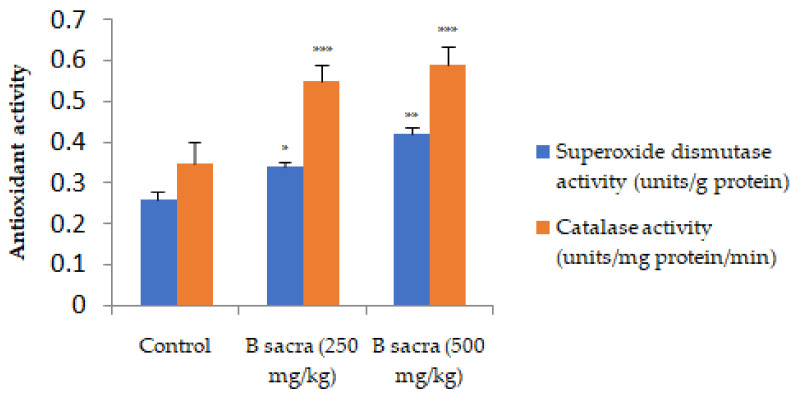
Effect on antioxidant enzymes. The SOD and catalase activities were determined in the testis by spectrophotometric method at the end of the treatment period. Statistical analysis was done using one-way ANOVA followed by Tukey’s test. All values are mean ± SEM, *n* = 6. * *p* < 0.05, ** *p* < 0.01, *** *p* < 0.001 compared to control.

**Figure 7 molecules-27-04699-f007:**
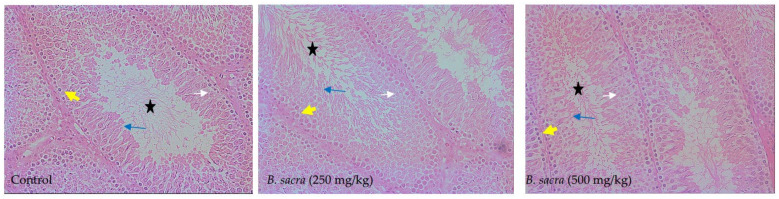
Histology of the testis of rats after different treatments (400×). Well-formed seminiferous tubules without any edema can be seen in the testis from animals of all the groups. Primary spermatocytes (yellow arrow), secondary spermatocytes (white arrow), spermatids (blue arrow), and sperms (black star) are seen.

**Figure 8 molecules-27-04699-f008:**
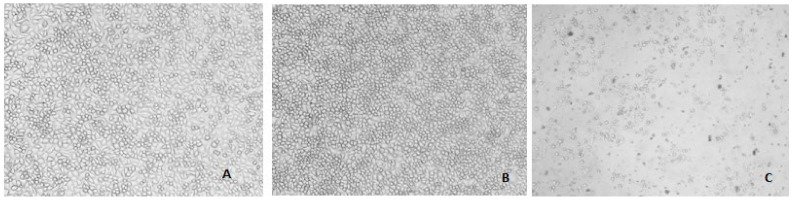
Photomicrographs showing human Leydig cells after incubation. (**A**) = untreated cells; (**B**) = *B. sacra* extract (100 µg/mL); (**C**) = LPS (10 µg/mL); the cell viability is more in the untreated cells and *B. sacra* extract-treated cells compared to cells treated with LPS.

**Figure 9 molecules-27-04699-f009:**
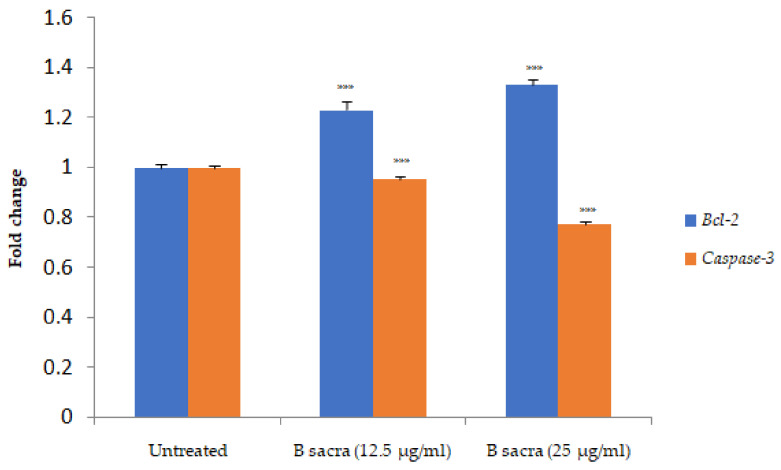
Effect on expression of *Bcl-2* and *caspase-3* in human Leydig cells (in vitro). The *Bcl-2* and *caspase-3* were measured in human Leydig cells after incubation with the *B. sacra* extract for 24 h. One-way ANOVA followed by Tukey’s test was used to analyze statistical significance. All values are mean ± SEM, *n* = 6, *** *p* < 0.001 compared to untreated cells.

**Table 1 molecules-27-04699-t001:** List of major constituents detected by GC-MS.

Number	Chemical Constituent	Retention Time (Rt)	Area (%)
1	1,2-Cyclooctanedione	4.529	2.30
2	2-Hydroxy-gamma-butyrolactone	5.861	2.20
3	PROCEROSIDE	6.883	2.15
4	1,4-Dioxin, 2,3-dihydro\-5,6-dimethyl	7.069	2.06
5	5-Hydroxymethylfurfural	11.856	2.25
6	2(4H)-Benzofuranone, 5,6,7,7a-tetrahydro-4,4	19.230	2.03
7	3-Ethoxy-4-hydroxyphenylacetonitrile	23.743	9.57
8	Pentadecanoic Acid	27.101	6.70
9	beta-d-Mannofuranoside, 1-*O*-(10-undeceny2)-	l8.012	3.98
10	9,12,15-Octadecatrienoic acid, (*Z*,*Z*,*Z*)-	29.346	6.80
11	Octadecanoic acid	29.595	2.13
12	Ethyl 1-thio-alpha-l-arabinofuranoside	33.310	3.25

**Table 2 molecules-27-04699-t002:** Cell viability assay on human Leydig cells (in vitro).

Treatment	Concentration (µg/mL)	Cell Viability (%) (Mean ± SEM)
Untreated		100
*B. sacra* extract	6.25	99.88 ± 1.23
	12.5	99.64 ± 0.98
	25	99.34 ± 2.35
	50	98.98 ± 0.89
	100	98.33 ± 0.32
LPS	0.5	89.9 ± 1.23
	1	73.83 ± 0.65 ^***^
	3	47.79 ± 0.44 ^***^
	6	33.93 ± 0.59 ^***^
	10	21.15 ± 0.36 ^***^

All values are mean ± SEM, *n* = 6. *** *p* < 0.001 compared to LPS (0.5 µg/mL).

**Table 3 molecules-27-04699-t003:** Primer sequences.

Gene	Primer Sequence	Accession Number
*Bcl-2*	5′ CATGTGTGTGGAGAGCGTCAAC 3′ (Forward primer)	NM_000633
	5′ CAGATAGGCACCCAGGGTGAT 3′ (reverse primer)	NM_000633
*Caspase-3*	5′ TATGGTTTTGTGATGTTTGTCC 3′ (Forward primer)	NM_001354783
	5′ TAGATCCAGGGGCATTGTAG 3′ (reverse primer)	NM_001354783
*GAPDH*	5′ TGACAACTTTGGTATCGTGGAAG 3′ (Forward primer)	NM_001357943
	5′ CAGTAGAGGCAGGGATGATGTT 3′ (reverse primer)	NM_001357943

## Data Availability

The raw data will be provided on request.

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
