# Peer review of "Beneficial Effect of Methanolic Extract of Frankincense (Boswellia Sacra) on Testis Mediated through Suppression of Oxidative Stress and Apoptosis"

_molecules, 2022, doi:10.3390/molecules27154699_

Round 1

Reviewer 1 Report

In the revised version of the manuscript, the authors significantly improved the content of the work. It is now clear and the results support the conclusions. However, a few aspects should be clarified:

In this work authors used rats aged 28 to 30 days that aren't sexually mature. Wistar rats reach sexual maturation at +/- 60 days after birth. This is relevant referring to the work. Additionally, it is not entirely clear why the authors chose concentrations of 250 mg/kg/day and 500 mg/kg/day. What is the rationale for that? Could you please explain? 

Could you please add in the table the acession number for the primers?

Author Response

In this work authors used rats aged 28 to 30 days that aren't sexually mature. Wistar rats reach sexual maturation at +/- 60 days after birth. This is relevant referring to the work. Additionally, it is not entirely clear why the authors chose concentrations of 250 mg/kg/day and 500 mg/kg/day. What is the rationale for that? Could you please explain? 

An explanation about the age of the animals has been given in the discussion along with reference as “Young animals aged 28 to 30 days were selected as it is known that agents targeting the male reproductive system will show a more profound effect if the administration is started at a young age and continued for a long duration[18]” (Page 9 para 3)

A further explanation is also given in the discussion (page 10, para 3) as “The current study was done using young animals as the extract had shown effect in adult rats in our previous study after 28-day repeated administration [9]. Since young animals were used, it can be suggested that the extract might have also influenced the development of the reproductive system. Further studies on adult rats after 60-days repeated administration in rats and the comparison of results from young animals with adult animals may help in understanding the effect of B. sacra extract on reproductive development. The in-vivo study was done using only two dose levels. Evaluation of effect at a wide range of doses will help in determining the threshold dose and maximum effective dose of the extract”.

The doses were selected based on an earlier study. It is now given in the discussion (lines 198 to 202)  as “The doses of the extract were selected based on an acute toxicity study carried in our earlier studies [8,9]. The extract administered orally was safe up to a dose of 2000 mg/kg and based on this, moderate doses of 250 mg/kg and 500 mg/kg were selected. The acute toxicity study was carried out as per the OECD guidelines following the Up and Down method.[19]”.

Could you please add in the table the accession number for the primers?

It is given now.

Reviewer 2 Report

-       The abstract of the paper highlights the essential contents (introduction, justification, general objective, material and methods, results, and main conclusion).

-       The Introduction contains all the information necessary to support the work.

-       I recommend the authors to give a brief description of the protocols described by Link, E.M. (1988) and Elstner, E.F. (1976) for the determination of superoxide dismutase (SOD) and catalase activities, respectively.

-       I recommend that the authors provide a more detailed description of the statistical analyses performed. What was the statistical significance established? what statistical software did they use? how did they evaluate the distribution of the data?

-       Using the standard error of the mean (SEM) as a measure of precision is a statistical error. The mean and standard error of the mean (SEM) are an estimate (the mean) and a measure of its precision (the SEM) for a characteristic of a population. The mean and standard deviation are the preferred summary statistics for (normally distributed) data.

-       A description of the statistical analysis of the data in figure 5 is missing.

-       The introductory paragraph of the discussion is too long. I recommend the authors to make a shorter introduction to the discussion and make clear the rationale for the study.

-       I recommend the authors to make a more detailed discussion of the effect of the extract on the enzymatic activity and expression level of bcl-2 and caspase-3 genes.

-       Considering the interesting results, how do the authors explain the effect of the extract on antioxidant enzyme activity and the level of expression of Bcl-2 and caspase-3 genes?

-       The authors state the following “Rats were used as experimental animals to determine the effect of Boswellia to extrapolate the same to humans because initial evaluations directly on humans are not possible due to ethical issues”. Extrapolation of data between different species (rats and humans) is a methodological error. I recommend the authors to remove this sentence from the manuscript. A solid discussion should not only leave readers with a deeper understanding of the topic, but also convince them that the current study has contributed to this area of knowledge.

Author Response

  • The abstract of the paper highlights the essential contents (introduction, justification, general objective, material and methods, results, and main conclusion).

 We appreciate reviewer’s observation

-       The Introduction contains all the information necessary to support the work.

We are thankful to you to the reviewer for his observation

-      I recommend the authors to give a brief description of the protocols described by Link, E.M. (1988) and Elstner, E.F. (1976) for the determination of superoxide dismutase (SOD) and catalase activities, respectively.

It is given now as per the suggestion of the reviewer

-       I recommend that the authors provide a more detailed description of the statistical analyses performed. What was the statistical significance established? what statistical software did they use? how did they evaluate the distribution of the data?

The level of significance and the software used in the analysis is given now. The distribution of data was assessed by Skewness value. A value between -1 to +1 was used to indicate normal distribution and a reference has been cited for the same [52].

-       Using the standard error of the mean (SEM) as a measure of precision is a statistical error. The mean and standard error of the mean (SEM) are an estimate (the mean) and a measure of its precision (the SEM) for a characteristic of a population. The mean and standard deviation are the preferred summary statistics for (normally distributed) data.

The SEM is used in almost all the animal studies. We referred to several publications from various journals including papers published in “Molecules” on animal studies. Almost all the research papers used SEM and not SD. Hence, this has not been changed. We also studied the below mentioned paper to understand whether SD or SEM is appropriate, we are convinced that for the data generated in the current study, SEM is more appropriate than SD. However, if the reviewer still insists that SD has to be given, we are ready to change SEM to SD.

Altman DG, Bland JM. Standard deviations and standard errors. BMJ. 2005 Oct 15;331(7521):903. doi: 10.1136/bmj.331.7521.903. PMID: 16223828; PMCID: PMC1255808. 

-       A description of the statistical analysis of the data in figure 5 is missing.

 It is given now

-     The introductory paragraph of the discussion is too long. I recommend the authors to make a shorter introduction to the discussion and make clear the rationale for the study.

The introductory paragraph is shortened now and the rationale of the study is given as per the suggestion of the reviewer

-       I recommend the authors to make a more detailed discussion of the effect of the extract on the enzymatic activity and expression level of bcl-2 and caspase-3 genes.

      We are thankful to the reviewer for his suggestion that is very useful to improve the quality of the manuscript. An explanation is now given in the discussion as “An increase in the expression of Bcl-2 and a reduction in caspase-3 expression might have contributed to an increase in catalase activity observed in-vivo in the rat testes. Bcl-2 is known to decrease hydrogen peroxide formation while caspase-3 is reported to increase hydrogen peroxide levels. The increase in catalase activity could be due to reduced hydrogen peroxide levels [38]. However, earlier reports suggest that Bcl-2 does not change activity of SOD [39]while an increase in catalase activity associated with a reduced expression of caspase-3 has been reported in many tissues [40,41]. On the contrary, administration of exogenous SOD is reported to induce expression of caspase-3 in prostate cancer cells lines [42]”.

-      Considering the interesting results, how do the authors explain the effect of the extract on antioxidant enzyme activity and the level of expression of Bcl-2 and caspase-3 genes?

An explanation is given above.

-      The authors state the following “Rats were used as experimental animals to determine the effect of Boswellia to extrapolate the same to humans because initial evaluations directly on humans are not possible due to ethical issues”. Extrapolation of data between different species (rats and humans) is a methodological error. I recommend the authors to remove this sentence from the manuscript. A solid discussion should not only leave readers with a deeper understanding of the topic, but also convince them that the current study has contributed to this area of knowledge.

The sentence is now deleted as per the reviewer’s suggestion

Reviewer 3 Report

In the manuscript “Beneficial effect of methanolic extract of frankincense (Boswellia sacra) on testis mediated through suppression of oxidative stress and apoptosis” the authors investigated the effect of methanolic extract of frankincense on the male reproductive system. They performed an in-vivo 60-day repeated oral administration study to determine the effect of B. sacra on sperms, testes, antioxidant enzymes, and serum hormonal levels in rats along with an in-vitro study on the human Leydig to determine cytotoxic effect and expression of apoptotic genes.

The topic is of significant relevance, so this manuscript could provide important reference information that are of great interest.

The manuscript is well written, the experimental design appears remarkable and is clearly illustrated. The figures are appropriate, in particular the figure 7. In addition, the authors have consulted a discrete number of scientific papers (50).

Author Response

In the manuscript “Beneficial effect of methanolic extract of frankincense (Boswellia sacra) on testis mediated through suppression of oxidative stress and apoptosis” the authors investigated the effect of methanolic extract of frankincense on the male reproductive system. They performed an in-vivo 60-day repeated oral administration study to determine the effect of B. sacra on sperms, testes, antioxidant enzymes, and serum hormonal levels in rats along with an in-vitro study on the human Leydig to determine cytotoxic effect and expression of apoptotic genes.

The topic is of significant relevance, so this manuscript could provide important reference information that are of great interest.

The manuscript is well written, the experimental design appears remarkable and is clearly illustrated. The figures are appropriate, in particular the figure 7. In addition, the authors have consulted a discrete number of scientific papers (50).

We are thankful to the reviewers for his observations and appreciation.

This manuscript is a resubmission of an earlier submission. The following is a list of the peer review reports and author responses from that submission.

Round 1

Reviewer 1 Report

In this study, the authors investigated the beneficial effects of Boswellia scare on male fertility. The authors begin their introduction (in the abstract section) by citing that Boswellia scare has been in use for the treatment of male infertility. However, in their methods they have conducted the study on normal rats, and towards the end they conclude that B.scara increases male fertility. I have the following observations for the authors;

  1. If this plant is known for its beneficial effect in male infertility, the authors should study it in a model of infertility.
  2. In such a case, the authors should conclude about its impact of alleviation of infertility rather than improvement in the fertility of normal rats. 
  3. The first line of the abstract and the last line do not go together. 
  4. Why did the authors prefer to use normal rats instead of an infertility model?
  5. The authors have done HPLC analysis of the extract, which is good. 
  6. The Leydig cell culture, was there a change in the secretion of testosterone? Were the cells stimulated or unstimulated?
  7. Anabolic effect on normal testis may not be so desirable unless it works in an infertility mode. 

Author Response

  1. If this plant is known for its beneficial effect in male infertility, the authors should study it in a model of infertility.

We appreciate the comments of the reviewer. We are in the process of planning the effect of Boswellia on cisplatin or cyclophosphamide induced testicular toxicity in rats. We have now mentioned this in the discussion (line 234 to 237). As we have mentioned in the discussion (line 158) that this study is a continuation of our earlier study where we had observed some beneficial effect of Boswellia in male rats and that result of the current study is a confirmation that Boswellia is beneficial for male fertility. Furthermore, there are several studies that have been carried out to report beneficial effect of different compounds on male fertility without using the infertility models. Some of them are given below

  1. Adelakun, S. A., Ogunlade, B., Olawuyi, T. S., & Ojewale, A. O. (2021). Aqueous extract of Tetrapleura tetraptera fruit peels influence copulatory behavior and maintain testicular integrity in sexually mature male Sprague-Dawley rats: Pro-fertility evaluation and histomorphometry evidence. Current research in physiology4, 7–16. https://doi.org/10.1016/j.crphys.2021.01.001
  2. Opuwari, C., & Monsees, T. (2020). Green tea consumption increases sperm concentration and viability in male rats and is safe for reproductive, liver and kidney health. Scientific reports10(1), 15269. https://doi.org/10.1038/s41598-020-72319-6
  3. Adeneye, A. A., Olagunju, J. A., & Murtala, B. A. (2019). Evaluation of Male Fertility-Enhancing Activities of Water Seed Extract of Hunteria umbellatain Wistar Rats. Evidence-based complementary and alternative medicine : eCAM2019, 7693010. https://doi.org/10.1155/2019/7693010

  1. In such a case, the authors should conclude about its impact of alleviation of infertility rather than improvement in the fertility of normal rats. 

As mentioned above, a study on the effect of Boswellia on infertility will be undertaken soon to further confirm its effect on male reproductive system. Since, infertility models induce severe damage to testis and it has not been evaluated whether Boswellia can attenuate such damage, it would be inappropriate from the findings of our study to conclude that Boswellia will alleviate infertility. Hence, no such conclusion was drawn.

  1. The first line of the abstract and the last line do not go together. 

The sentence is now modified as per the reviewer’s suggestion

  1. Why did the authors prefer to use normal rats instead of an infertility model?

As mentioned above, this study was a continuation of our previous study where a beneficial effect with few parameters was observed after a 28-day repeated dose administration. Hence, the same was confirmed after 60 day administration in young animals using more parameters and also an ­in-vitro study was also carried out in the current study to further confirm its effect

  1. The authors have done HPLC analysis of the extract, which is good. 

We are thankful to the reviewer for his appreciation of HPTLC analysis

  1. The Leydig cell culture, was there a change in the secretion of testosterone? Were the cells stimulated or unstimulated?

The testosterone levels were not measured in the Leydig cells and the cells were unstimulated. The study was done to determine the effect on genes involved in apoptosis. This is now mentioned in the discussion (Line 228).

  1. Anabolic effect on normal testis may not be so desirable unless it works in an infertility mode

We agree with the reviewer. However, the conclusion of our results only claims beneficial effect on male fertility and it does not claim that this herb will treat infertility. As mentioned above, we have now given a sentence in the discussion that evaluation of effect on infertility models both in-vivo and in-vitro is required to further confirm its effect on infertility.

Reviewer 2 Report

The present manuscript by SA Alharbi et al. deals with the description of a beneficial effect (attributed to anti-oxidative action) of frankincense methanolic extract on rat testis in vivo and on human Leydig cells in vitro.

General comment
I regret to say that the study presented in this paper has low scientific interest and is not very original. Moreover, several experiments are not convincing because of drawbacks or lack of information.

Specific comments

  1. The title should include « methanolic » extract as the authors indicate in the introduction that aqueous extract has completely different effects.
  2. Table 1 is not very informative without any numeric data, and a simple sentence would have been sufficient. Or the data mentioned at lines 63-70 should be included in table I.
  3. Figure 1 is not easy to read and, nevertheless, incomplete as it does not permit the identification of the different components.
  4. In figures 2-5 (in vivo experiments in rats), only two close extract concentrations were tested. A wider range of concentrations would be much more informative (effect threshold, saturation, etc..)
  5. In figure 3, the identity of the grey boxes is not given. It is LH level, obviously, but it should be indicated. The FSH and LH concentrations are given in ng/l (i.e., pg/ml). This assay is thus extraordinarly sensitive. It is a pity that the technique used is not described !!
  6. In all figures, the captions should be extended to give more precise and readable experimental details.
  7. Lines 160 : the animals were included in the protocols starting at 28-30 days of age. These animals are still immature and just started their puberty. The observed effects are instead due to interference with the development of testicular function than testicular function itself.
  8. Line 180 : some of the antioxidants previously shown to prevent testicular damage should be included in the experiments of the present paper for the sake of comparison.
  9. Lines189-192 : this sentence is not clear (The conclusion is not logic).
  10. Lines 237 : Precise suppliers
  11. Line 273 : It is indicated that the rats are « treated » with the extract but not through which route (IV or IM injection, or gavage?
  12. Line 274 : indicate that the doses are « mg/kg/day » for 60 days.
  13. Line 315 : it is unclear why the authors used human instead of rat Leydig cells. It would have been more coherent to use the same species in in-vivo and in-vitro experiments.

Author Response

  1. The title should include « methanolic » extract as the authors indicate in the introduction that aqueous extract has completely different effects.

It is now done as per the suggestion of the reviewer

  1. Table 1 is not very informative without any numeric data, and a simple sentence would have been sufficient. Or the data mentioned at lines 63-70 should be included in table I.

The table 1 is now deleted and results are described in the text

  1. Figure 1 is not easy to read and, nevertheless, incomplete as it does not permit the identification of the different components.

As mentioned in the text, we have given HPTLC ‘fingerprint’ in the manuscript. “A HPTLC fingerprint is used for identification of species and in some cases to certify the geographical origin of the species. It is considered as one of the best methods for one step validation of quality, purity, stability and identity”. It is now mentioned in the manuscript under discussion (Line 152-156).

  1. In figures 2-5 (in vivo experiments in rats), only two close extract concentrations were tested. A wider range of concentrations would be much more informative (effect threshold, saturation, etc..)

The present study was done to evaluate the effect of Boswellia extract on the reproductive system and to determine if the effect is dose dependent. No attempt was made to determine the threshold dose or maximum effective dose as it is beyond the scope of the study. The doses were selected based on earlier studies. Two dose levels are considered enough to determine dose dependent effect. Some of recent publications where only two doses were used are given below

  1. Paul, A. K., Gueven, N., & Dietis, N. (2021). Profiling the Effects of Repetitive Morphine Administration on Motor Behavior in Rats. Molecules (Basel, Switzerland)26(14), 4355. https://doi.org/10.3390/molecules26144355
  2. Saadullah, M., Asif, M., Farid, A., Naseem, F., Rashid, S. A., Ghazanfar, S., Muzammal, M., Ahmad, S., Bin Jardan, Y. A., Alshaya, H., Saleem, M. H., Ali, S., Adetunji, C. O., & Arif, S. (2022). A Novel Distachionate from Breynia distachiaTreats Inflammations by Modulating COX-2 and Inflammatory Cytokines in Rat Liver Tissue. Molecules (Basel, Switzerland)27(8), 2596. https://doi.org/10.3390/molecules27082596
  3. Alyahya, A., & Asad, M. (2020). Repeated 28-DAY oral dose study on Boswellia sacra oleo gum resin extract for testicular toxicity in rats. Journal of ethnopharmacology258, 112890. https://doi.org/10.1016/j.jep.2020.112890
  4. Al-Yahya, A., Asad, M., Sadaby, A., & Alhussaini, M. S. (2020). Repeat oral dose safety study of standardized methanolic extract of Boswellia sacra oleo gum resin in rats. Saudi journal of biological sciences27(1), 117–123. https://doi.org/10.1016/j.sjbs.2019.05.010

  1. In figure 3, the identity of the grey boxes is not given. It is LH level, obviously, but it should be indicated. The FSH and LH concentrations are given in ng/l (i.e., pg/ml). This assay is thus extraordinarly sensitive. It is a pity that the technique used is not described !!

The LH box was not seen due to formatting error. It can be seen now. The estimation of serum hormone levels was carried out using standard methods that has been described by several authors. Hence, only references were provided.

  1. In all figures, the captions should be extended to give more precise and readable experimental details.

The captions are explanatory and it shows statistical difference between the groups. It is a standard practice to provide captions in this way. We apologize but we could not understand what else needs to included.

  1. Lines 160 : the animals were included in the protocols starting at 28-30 days of age. These animals are still immature and just started their puberty. The observed effects are instead due to interference with the development of testicular function than testicular function itself.

We have already discussed why animals of this age were used with a suitable reference (line 165-171). It is given as “Young animals aged 28 to 30 days were selected as it is known that agents targeting male reproductive system will show more profound effect if administration is started at a young age and continued for long duration[19].” Interference in the development of testicular function usually results in reduced levels of testosterone and less testicular function. However, such an effect was not observed in the current study. Furthermore, we have mentioned in the manuscript that this study is a continuation of our study wherein beneficial effect was observed after 28-day repeated administration to adult rats but only few parameters were studied. This study is more comprehensive and is a confirmation of beneficial effect of B. sacra on male fertility.

  1. Line 180 : some of the antioxidants previously shown to prevent testicular damage should be included in the experiments of the present paper for the sake of comparison.

Antioxidant effect may one the mechanisms by which the extract has shown effect. We have given this in the conclusion as “B. sacra (frankincense) extract showed potential protective effect in the testis that is mediated at least in part due to its antioxidant action”. Most of the studies reporting fertility promoting effect have described antioxidant as one of the mechanisms and none of them used standard antioxidant for comparison. Some of the studies are given below

  1. Adelakun, S. A., Ogunlade, B., Olawuyi, T. S., & Ojewale, A. O. (2021). Aqueous extract of Tetrapleura tetraptera fruit peels influence copulatory behavior and maintain testicular integrity in sexually mature male Sprague-Dawley rats: Pro-fertility evaluation and histomorphometry evidence. Current research in physiology4, 7–16. https://doi.org/10.1016/j.crphys.2021.01.001
  2. Opuwari, C., & Monsees, T. (2020). Green tea consumption increases sperm concentration and viability in male rats and is safe for reproductive, liver and kidney health. Scientific reports10(1), 15269. https://doi.org/10.1038/s41598-020-72319-6
  3. Adeneye, A. A., Olagunju, J. A., & Murtala, B. A. (2019). Evaluation of Male Fertility-Enhancing Activities of Water Seed Extract of Hunteria umbellatain Wistar Rats. Evidence-based complementary and alternative medicine : eCAM2019, 7693010. https://doi.org/10.1155/2019/7693010
  1. Lines189-192 : this sentence is not clear (The conclusion is not logic).

We apologize for formatting errors. Lines 189-192 describes “antioxidants are also recommended for the treatment of infertility to protect testis from endogenous oxidants and it has been reported that consumption of antioxidant supplements by men may increase live birth rate in infertile couples[26]” and it does not describe conclusion” We would appreciate if the reviewer describes the sentence that needs to modified.

  1. Lines 237 : Precise suppliers

Since a lot of chemicals were used for biochemical, genetic and pharmacological evaluations from different suppliers, a sentence “The reagents, chemicals and diagnostic kits were procured from different suppliers” was mentioned in the methodology

  1. Line 273 : It is indicated that the rats are « treated » with the extract but not through which route (IV or IM injection, or gavage?

We have already mentioned in the methodology that extract was administered orally (Line 286). “The second group and third group were treated with a suspension of methanolic extract of B. sacra orally at a dose of 250 mg/kg and 500 mg/kg for 60 days respectively[8].”

  1. Line 274 : indicate that the doses are « mg/kg/day » for 60 days.

It is modified as per reviewer’s suggestion as “The second group and third group were treated with a suspension of methanolic extract of B. sacra orally at a dose of 250 mg/kg/day and 500 mg/kg/day for 60 days respectively[8].”

  1. Line 315 : it is unclear why the authors used human instead of rat Leydig cells. It would have been more coherent to use the same species in in-vivo and in-vitro experiments.

“Rats were used as experimental animals to determine the effect of Boswellia with an aim to extrapolate the same to humans because initial evaluations directly on humans are not possible due to ethical issues. However, such hinderance is not seen with cell lines from humans. Hence, human cell lines were used for in-vitro studies and in-vivo experiments were done in rats”. The same is now mentioned in the discussion (line 241 to 245).

Reviewer 3 Report

This study on the effects of frankincense on male fertility is well designed, with robust methodology, well-presented results, and a very thorough discussion.

The manuscript needs minor corrections and additions before it is accepted.

Minor

line 106: “…figure 8.”, change “8” with “6”

lines 108-110: “…were more in ........... compared to control.”, add statistic

line 121: “(Figure  7)”, change “(Table 2 and Figure 7)”

Table 2. add statistic of LPS data

Figure 7: invert photo B (LPS) with photo C (B. sacra extract)

Author Response

  1. line 106: “…figure 8.”, change “8” with “6”

Appreciate the comments. It is done now

  1. lines 108-110: “…were more in ........... compared to control.”, add statistic

No statistical test was done as it was visual evaluation. Hence, it was not mentioned.

  1. line 121: “(Figure  7)”, change “(Table 2 and Figure 7)”

It is done now

  1. Table 2. add statistic of LPS data

It is given now. Since comparison cannot be made with untreated as SEM is 0. Comparison was made with lowest concentration. It was non-significant in case of B. sacra extract but significant with LPS

  1. Figure 7: invert photo B (LPS) with photo C (B. sacra extract)

It is done now

Round 2

Reviewer 1 Report

The authors have adequately addressed my comments. 

Author Response

We are thankful to the reviewer for accepting the revision. 

Reviewer 2 Report

This new version has been very quickly resubmitted following the reviewers' comments on version 1.

Concerning my own comments, they have been only marginally taken into consideration and version 2 of the manuscript is still far from satisfactory. The authors have chosen to reply rapidly without really improving the manuscript. For example, they did not accept to mention the various references and suppliers for the different chemicals they used and did not accept to complete the figures’ captions as I required. The figures that are difficult to read and not informative enough have not been changed.

The fundamental issues I raised were not discussed in-depth or even not taken into consideration for complementary experiments (age of animals, concentration range, etc.).

Author Response

  1. Concerning my own comments, they have been only marginally taken into consideration and version 2 of the manuscript is still far from satisfactory. The authors have chosen to reply rapidly without really improving the manuscript. For example, they did not accept to mention the various references and suppliers for the different chemicals they used and did not accept to complete the figures’ captions as I required. The figures that are difficult to read and not informative enough have not been changed.

We regret that we could not completely answer or modify the manuscript as per the reviewer’s suggestions

We have now cited the suppliers wherever we have mentioned the name of chemicals in the manuscript. We had already mentioned about the source of cell lines and the source of chemicals for gene expression analysis. We have now mentioned source of other chemicals. For estimations such as antioxidant enzyme determinations and ELISA methods, we have not mentioned the procedure in detail as standard methods commonly reported by different authors were used. However, if the reviewer requires that all the procedures be mentioned in detail along with chemicals and their suppliers to improve the quality of the manuscript, we would do it.

The figure captions have now been given by referring to science editing website (https://www.internationalscienceediting.com/how-to-write-a-figure-caption/) consisting of declarative title, methods and statistical information. However, we would add more captions if the reviewer suggests specifically the corrections required.

In the earlier revision, the reviewer had written that figure 1 is not easy to read and does not permit identification of the compounds. We have modified the figure and have given the pictures of HPTLC plates similar to earlier publications mentioning the HPTLC analysis of different plant extract. Five flavonoids and one tannin were detected in the HPTLC fingerprint. As mentioned in our previous reply, “A HPTLC fingerprint is used for identification of species and in some cases to certify the geographical origin of the species. It is considered as one of the best methods for one step validation of quality, purity, stability and identity”. It was in the first revision now mentioned in the manuscript under discussion (Line 152-156). We have cited references below to show that the presentation of HPTLC results in this manuscript is similar to earlier published papers. However, if the reviewer feels that the figures have to be further modified, we would welcome it with specific correction that needs to be done.

  1. Chothani, D. L., Patel, M. B., & Mishra, S. H. (2012). HPTLC fingerprint profile and isolation of marker compound of Ruellia tuberosa. Chromatography Research International2012.
  2. Pudumo, J., Chaudhary, S. K., Chen, W., Viljoen, A., Vermaak, I., & Veale, C. G. L. (2018). HPTLC fingerprinting of Croton gratissimus leaf extract with Preparative HPLC-MS-isolated marker compounds. South African Journal of Botany114, 32-36.
  3. Karthika, K., & Paulsamy, S. (2015). TLC and HPTLC fingerprints of various secondary metabolites in the stem of the traditional medicinal climber, solena amplexicaulis. Indian journal of pharmaceutical sciences77(1), 111.
  4. Saraf, A. Y., & Saraf, A. A. (2020). HPTLC Fingerprinting: A Tool for Simplified Analysis of Phenolics in Medicinal Plants. INDIAN JOURNAL OF PHARMACEUTICAL EDUCATION AND RESEARCH54(4), 1098-1103.
  5. Seboletswe, P. S., Mkhize, Z., & Katata-Seru, L. M. (2019). HPTLC Fingerprint Profiling of Protorhus longifolia Methanolic Leaf Extract and Qualitative Analysis of Common Biomarkers. International Journal of Materials and Metallurgical Engineering13(12), 553-557.
  6. The fundamental issues I raised were not discussed in-depth or even not taken into consideration for complementary experiments (age of animals, concentration range, etc.).

As per the suggestion of the reviewer, we have now discussed the issues raised about the age of the animals. We have now written that the extract might also have affected the development of reproductive system because the extract was administered to young animals. We have also given a statement to mention that the current study was done using only two doses. Further studies have to be carried out to using different doses to determine the threshold dose and the maximum effective dose of the extract.

The explanations are given as “The current study was done using young animals as the extract had shown effect in adult rats in our previous study after 28-day repeated administration [9]. Since, young animals were used, it can be suggested that the extract might have also influenced the development of reproductive system. Further studies on adult rats after 60-days repeated administration and the comparison of results from young animals with adult animals may help in understanding the effect of B. sacra extract on reproductive development. The in-vivo study was done using only two dose levels. Evaluation of effect at a wide range of doses will help in determining the threshold dose and maximum effective dose of the extract’ (Page 16, para 1).